# Green Space Optimization Strategy to Prevent Urban Flood Risk in the City Centre of Wuhan

**Yajing Liu** [1,†], **Yan Zhou** [2,*,†], **Jianing Yu** [2], **Pengcheng Li** [3] and **Liuqi Yang** [2]

1  Introduction of Shaanxi Provincial Land Engineering Construction Group Co., Ltd., Xi'an 710000, China; liuyajing0218@foxmail.com
2  Department of Urban Planning, School of Urban Design, Wuhan University, Wuhan 430071, China; yjnyujianing@foxmail.com (J.Y.); 2019202090024@whu.edu.cn (L.Y.)
3  Department of Urban Planning, School of Architecture, Southeast University, Nanjing 210000, China; lipengcheng115230@163.com
*  Correspondence: joyeezhou@whu.edu.cn
†  Authors contributed equal.

**Abstract:** Changing the water permeability ratio of urban underlying surface helps alleviate urban flood. This paper designs the swale identification experiment to modify the flood-submerging simulation experiment based on the SCS-CN model and proves that the results generated by the modified experiment better reflect the realities. The modified flood-submerging simulation experiment is then applied to downtown Wuhan to obtain the quantitative data. The data are used to quantify the catchment capacities of the lots. Based on the rainfall collection capacities, the maximum surface rainfall runoff volume that would not cause flood is arrived at using the rainfall runoff formula. The maximum runoff volume represents the rainwater storage capacities of the lot based on the proportion of the green space that is identified within the study area. The results suggest that this rainwater storage capacity evaluation model works efficiently to identify the urban areas with flood risks and provides the rainwater runoff thresholds for different areas. Adjustments in the spatial patterns and proportions of the green space help ensure that the rainwater runoff volume is below the thresholds, thus contributing to the prevention and control of the urban flood risks.

**Keywords:** urban flood; hydrologic modeling; rainwater storage; land-use planning; green coverage; low-impact development; urban hydrologic processes



## 1. Introduction

Urban flood disasters resulting from heavy rainfall are on the increase over the past few years, causing enormous losses to most of China's cities. One of the root causes of this problem is the rapid and undifferentiated expansion of the impermeable urban surfaces due to urbanization. The process significantly changes the catchment hydrology, increasing runoff rates and volumes on one hand and undermining infiltration and baseflow (provided there is no additional source of baseflow) on the other hand. The original hydrologic and ecological environment in urban areas is damaged as well [1–5].

Fundamentally, the essence of non-engineering measures to alleviate flood in urban areas aim twofold: (1) to restore the pre-development hydrologic patterns of a site by regulating the volumes and rates of the urban hydrologic processes and (2) to ensure the independent absorption of the rainfall within a basin so as to avoid flow concentration among basins [6–8]. The urban flood can be effectively avoided by regulating the hydrological process in an ecological way [9,10]. Therefore, the accurate simulation or prediction of storm runoff is one of the most important bases of water resource management [11].

Hydrologic models, as the basic tools to estimate the peak volumes and flood peaks [12], have been widely studied by scholars in the field. There are two types of hydrologic models today. One is the performance evaluation models and the stormwater-management models,

with complicated data such as soil infiltration and drainage network as the input. The simulation or prediction performance is subject to the high-precision survey of temporal-spatial patterns and the accurate prediction of the rainfall. However, it can be challenging to obtain such data for a city. Even if the data are obtained, they can hardly generate accurate simulation results. Therefore, these models are seldom adopted to predict the rainwater runoff within a large-scale space [13,14]. In addition, the output of these models is not included in the indicators of the urban comprehensive plans [15,16]. The other type is the GIS-based hydrologic scenario simulation widely used in large-scale urban planning. Models are established to calculate and simulate the submerged areas and water depths during different storm recurrence periods. Combined with typological, precipitation, and drainage models, these models generate the distribution patterns of the waterlogged areas [17–19]. The SCS-CN model is designed by the Soil Conservation Service under the United States Department of Agriculture (USDA-SCS) to simulate the flood peaks and runoff volumes of the floods brought about by extreme weather within a small basin. This ready-to-use runoff calculation model takes the relationships between runoff volumes and precipitation, soil types, land use and management conditions, weathers, and soil moistures into consideration. Empowered by GIS toolkits, the SCS-CN model is able to identify the areas that are vulnerable to flood, the submerging volumes, and the submerging depths within a city based on the results from the flood-submerging simulation experiments. The model is widely applied now [20,21]. However, some studies have shown that this method cannot effectively determine the impact of precipitation storage and consumption on runoff, which limits the accuracy of runoff prediction by this method [22,23]. At the same time, it is established based on the assumption that all rainwater submerges the lowest point gradually, which deviates from the actual rainfall-submerging scenes [24,25]. To make up for this deficiency, this study innovatively designed the swale identification experiment to improve the SCS-CN model.

Wuhan features a sub-tropical monsoon climate with a long rainy season, concentrated storms, large rainfall volumes, contemporaneous rainy season and flood season, and delayed flood peaks. The city is therefore more vulnerable to storm-related disasters. The occurrence of urban flood is a complex problem involving the integrated water system and also includes the failure of urban drainage systems [26,27]. However, the urban, centralized drainage pipe system with the single purpose of drainage cannot adapt to short-term rainstorms in most cities in China and will also cause damage to the urban water ecological environment. As a city by the lake and the river, Wuhan has a rich and complex urban water system, the complex urban water ecological problems of which cannot be solved only through single-objective grey engineering measures [28,29].

With downtown Wuhan as the study area, this paper modifies the SCS-CN-based flood-submerging simulation experiment, establishes the rainwater storage capacity evaluation model with mathematical analysis methods, and identifies the areas that can give the fullest play to the role of the infiltration and depression processes in Wuhan. The evaluation results are transformed into the planning-control indicators related to the green ratio, which help to ensure that the urban hydrologic patterns are not drastically undermined at the urban comprehensive planning stage and to boost the hydrologic-cycle efficiency across the region.

## 2. Materials and Methods

A total of 1530 lots in the Wuhan Comprehensive Plan 2006–2020 were selected. The GIS hydrological analysis tools were employed in the swale identification experiment and the flood-submerging simulation experiment. The catchment capacities of the lots were calculated based on the experiments [30]. The results were then used as the input of the comprehensive runoff coefficient calculation formula to assess the catchment capacity of the lots. The evaluation results were transformed into the minimal green coverage, which was applied in the land space planning. Please refer to Appendix A Tables A1–A4 and Figure A1 for the data and their source used in the experiments. The experimental

procedure is as follows; the detailed operation procedure is in the Supplementary Material Tables S1 and S2.

### 2.1. Catchment-Capacity Simulation Experiment

The catchment capacity refers to the water volume that can be accommodated by a catchment after the depression hydrological process without infiltration. This indicator was used to reflect the accumulated precipitation and the capacity to collect the rainwater from the surrounding area of the catchment. It can be expressed by the accumulated water volume per unit of projected area when a catchment is filled up with the rainwater. The formula is shown as below:

$$\beta = \frac{V}{S} \tag{1}$$

where $\beta$ represented the catchment capacity (m$^3$/ m$^2$), $V$ the maximum rainwater volume (m$^3$) a catchment can collect, and $S$ the projected area (m$^2$) of the catchment on the horizontal plane.

To calculate the catchment capacity, the target return period should be identified first. This paper calculated the catchment capacity during the 100-year return period (decomposed into 6 grades: namely 1-year, 5-year, 10-year, 20-year, 50-year, and 100-year return periods). The values of $\beta$ for different return periods were drawn based on the swale identification experiment and the flood-submerging simulation experiment.

### 2.1.1. Swale Identification Experiment

In this experiment, all swales were identified based on the topographic features. The difference between the overflow point (the highest point) and the bottom point (the lowest point) of each swale was arrived at to reflect the depth of the swale. The catchment capacity was calculated based on the swale depth.

This experiment was conducted in two steps. Step 1: Identified the swales. Processed the elevation data in Wuhan with the flow-direction tools to obtain the flow-direction data; used the hydrological confluence tools to identify all swales in Wuhan; and divided these swales into watersheds with the watershed tools. Step 2: Calculated the swale depth. Employed the regional analysis-zonal statistics tools to get the minimal elevation of each swale; used the regional analysis-region filling tools to identify the overflow point of each swale; and adopted the raster calculator to obtain the difference between the overflow point and the bottom point of each swale, i.e., the depth of the swale.

For a city, detailed threshold division in the swale-identification experiment would better reflect the catchment capacity of all lots within the study area. This experiment took the depression process into consideration only. The results reflected the relationship between the terrains and the hydrologic patterns and did not represent the actual rainwater storage capacity. The impact of elevation on the catchment capacity within small watersheds was not taken into account either.

### 2.1.2. SCS-CN-Based Flood-Submerging Simulation Experiment

The SCS-CN model was adopted to simulate the precipitation process and the flood-submerging area during various return periods, thus identifying the area with the largest catchment capacity.

This experiment was composed of three steps. Step 1: Divided the watersheds. Obtained the flow data in downtown Wuhan with the Flow Direction tools and the Flow tools. By setting the Flow Threshold for each watershed to be 1,500,000, the downtown area was divided into 24 watersheds. Step 2: Calculated the daily submerging volume during extreme conditions. Employed the SCS-CN model to calculate the submerging volume of each threshold during various return periods. Step 3: Calculated the flood-submerging elevation. Used the surface volume tools and the dichotomy (0.001) to estimate the flood-submerging elevation of all watersheds during various return periods. A detailed explanation of the dichotomy: The submergence elevation of each watershed in different

recurrence periods is estimated by using dichotomy method (accuracy 0.001). First, we can estimate a range of submergence height (a,c), then calculate the submergence volume of mid-point c, and then compare the result with that of the submergence volume of each basin; if it is small, the range of values of the elevation becomes (c,b). Repeat the above steps until the calculated elevation is infinitely close to the submerged volume of each watershed.

This experiment took precipitation and infiltration processes into consideration, examining the flood-submerged conditions under different rainfall. However, the catchment capacity of the regions other the flood-submerged areas was not simulated. After getting the rainfall, the rainwater was filled into the thresholds from the lowest point until they were submerged completely, which deviated from the actual precipitation process.

### 2.1.3. Catchment-Capacity Calculation

To sum up, the SCS-CN-based flood-submerging simulation experiment took more factors into consideration and better reflected the actual hydrologic process. The experiment principle is as shown in Figure 1. However, this experiment cannot reveal the catchment capacity without the aid of the swale identification experiment (Figure 2). Data from the swale identification experiment was further processed because it did not take the precipitation amount into consideration. For regions where the precipitation amount was larger than the swale volume, the catchment capacity was expressed as the ratio of the swale volume to the horizontal plane projection of the swale; for regions where the precipitation amount was smaller than the swale volume, the simulation experiment for the catchment was repeated before the catchment capacity was calculated. The fusion principle of two experiments was as shown in Figure 3. Based on the interpretation of the flooded area from the Wuhan satellite image data and the monitoring data of the relevant technical departments in Wuhan in recent years, the actual flooded area of Wuhan was obtained (Figure 4). Comparison between the experiment results and the actual flood-submerged conditions suggested that the fused results were more accurate.

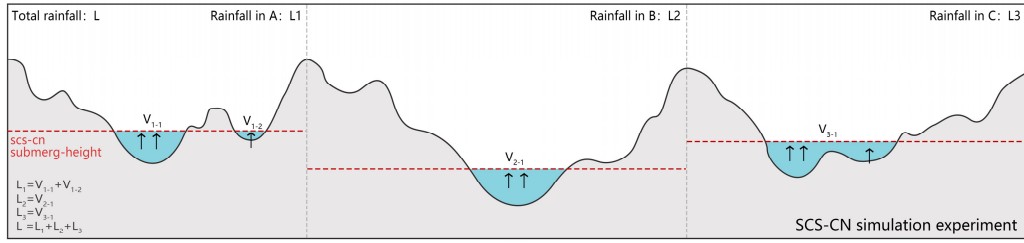

**Figure 1.** SCS-CN-based flood-submerging simulation experiment.

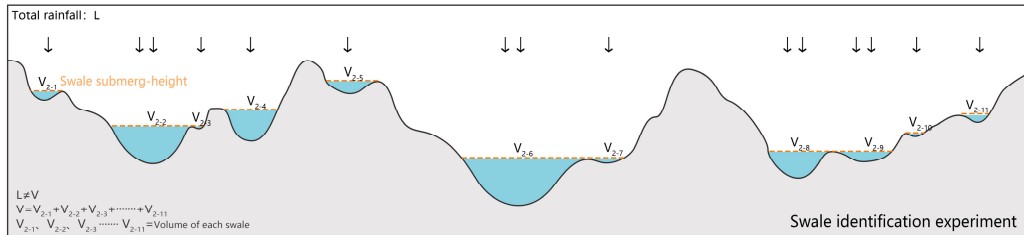

**Figure 2.** Swale identification experiment.

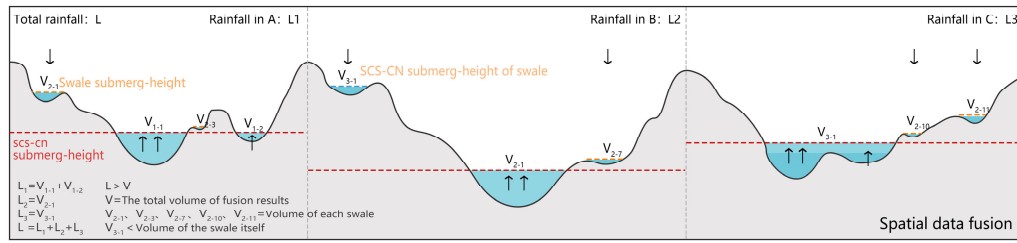

**Figure 3.** Spatial data fusion.

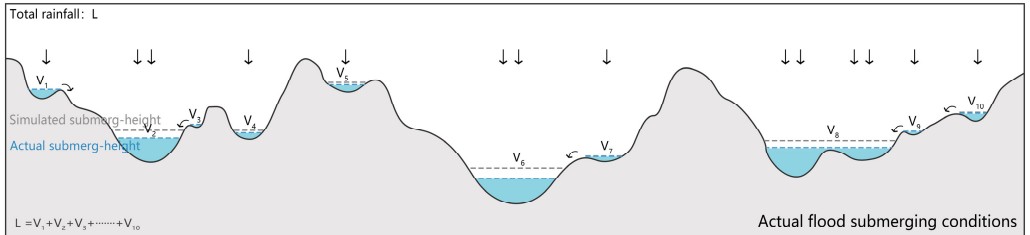

**Figure 4.** Actual flood-submerged conditions.

Therefore, the rainfall-collection capacity within the 100-year return period was calculated differently depending on the actual conditions (Figure 5).

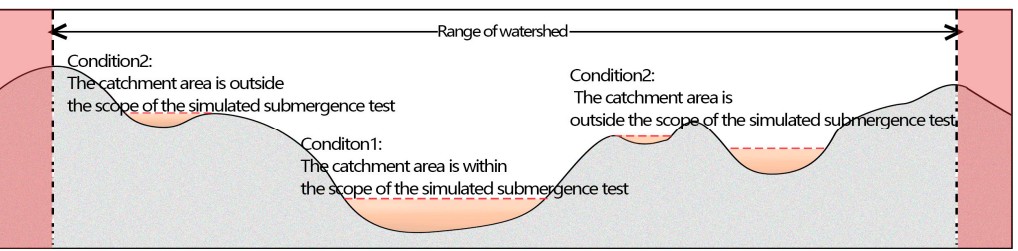

**Figure 5.** Submerged area location.

(i) When the catchment was within the simulated flood-submerged area, the catchment capacities during different return periods were calculated as below:

$$\beta_a = \frac{Q_a - Q_{a-t}}{1000} \times \frac{\frac{1}{2}S_a}{S_1 + \cdots\cdots + S_{a-t} + \frac{1}{2}S_a} + \frac{(Q_{a+t} - Q_a)}{1000} \times \frac{S_a}{S_1 + \cdots\cdots + S_{a-t} + S_a + \frac{1}{2}S_{a+1}} + \cdots\cdots + \frac{(Q_x - Q_{x-t})}{1000} \times \frac{S_a}{S_1 + \cdots\cdots + S_{x-t} + \frac{1}{2}S_x} \quad (2)$$

where $\beta_a$ represented the catchment capacity ($m^3/m^2$) of the simulated a-year flood-submerged area, $S_a$ the horizontal plane projection area ($m^2$) of the simulated a-year flood-submerged area, and $Q_a$ the simulated a-year rainfall (mm).

The simulated flood inundation area is different in different years (Figure 6).

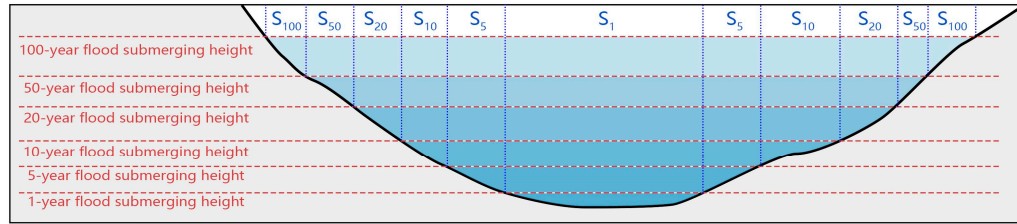

**Figure 6.** Calculation of catchment capacity within the simulated flood-submerged areas.

For example, calculation of catchment capacity for simulated 1-year flood-submerged area ($S_1$):

$$\beta_1 = \frac{Q_1}{1000} + \frac{(Q_5 - Q_1)}{1000} \times \frac{S_1}{S_1 + \frac{1}{2}S_5} + \frac{(Q_{10} - Q_5)}{1000} \times \frac{S_1}{S_1 + S_5 + \frac{1}{2}S_{10}} + \frac{(Q_{20} - Q_{10})}{1000} \times \frac{S_1}{S_1 + S_5 + S_{10} + \frac{1}{2}S_{20}} + \frac{(Q_{50} - Q_{20})}{1000} \times \frac{S_1}{S_1 + S_5 + S_{10} + S_{20} + \frac{1}{2}S_{50}} + \frac{(Q_{100} - Q_{50})}{1000} \times \frac{S_1}{S_1 + S_5 + S_{10} + S_{20} + S_{50} + \frac{1}{2}S_{100}} \tag{3}$$

Calculation of catchment capacity for simulated 5-year flood-submerged area ($S_5$):

$$\beta_5 = \frac{(Q_5 - Q_1)}{1000} \times \frac{\frac{1}{2}S_5}{S_1 + \frac{1}{2}S_5} + \frac{(Q_{10} - Q_5)}{1000} \times \frac{S_5}{S_1 + S_5 + \frac{1}{2}S_{10}} + \frac{(Q_{20} - Q_{10})}{1000} \times \frac{S_5}{S_1 + S_5 + S_{10} + \frac{1}{2}S_{20}} + \frac{(Q_{50} - Q_{20})}{1000} \times \frac{S_5}{S_1 + S_5 + S_{10} + S_{20} + \frac{1}{2}S_{50}} + \frac{(Q_{100} - Q_{50})}{1000} \times \frac{S_5}{S_1 + S_5 + S_{10} + S_{20} + S_{50} + \frac{1}{2}S_{100}} \tag{4}$$

Calculation of catchment capacity for simulated 10-year flood-submerged area ($S_{10}$):

$$\beta_{10} = \frac{(Q_{10} - Q_5)}{1000} \times \frac{\frac{1}{2}S_{10}}{S_1 + S_5 + \frac{1}{2}S_{10}} + \frac{(Q_{20} - Q_{10})}{1000} \times \frac{S_{10}}{S_1 + S_5 + S_{10} + \frac{1}{2}S_{20}} + \frac{(Q_{50} - Q_{20})}{1000} \times \frac{S_{10}}{S_1 + S_5 + S_{10} + S_{20} + \frac{1}{2}S_{50}} + \frac{(Q_{100} - Q_{50})}{1000} \times \frac{S_{10}}{S_1 + S_5 + S_{10} + S_{20} + S_{50} + \frac{1}{2}S_{100}} \tag{5}$$

Calculation of catchment capacity for simulated 20-year flood-submerged area ($S_{20}$):

$$\beta_{20} = \frac{(Q_{20} - Q_{10})}{1000} \times \frac{\frac{1}{2}S_{20}}{S_1 + S_5 + S_{10} + \frac{1}{2}S_{20}} + \frac{(Q_{50} - Q_{20})}{1000} \times \frac{S_{20}}{S_1 + S_5 + S_{10} + S_{20} + \frac{1}{2}S_{50}} + \frac{(Q_{100} - Q_{50})}{1000} \times \frac{S_{20}}{S_1 + S_5 + S_{10} + S_{20} + S_{50} + \frac{1}{2}S_{100}} \tag{6}$$

Calculation of catchment capacity for simulated 50-year flood-submerged area ($S_{50}$):

$$\beta_{50} = \frac{(Q_{50} - Q_{20})}{1000} \times \frac{\frac{1}{2}S_{50}}{S_1 + S_5 + S_{10} + S_{20} + \frac{1}{2}S_{50}} + \frac{(Q_{100} - Q_{50})}{1000} \times \frac{S_{50}}{S_1 + S_5 + S_{10} + S_{20} + S_{50} + \frac{1}{2}S_{100}} \tag{7}$$

Calculation of catchment capacity for simulated 100-year flood-submerged area ($S_{100}$):

$$\beta_{100} = \frac{(Q_{100} - Q_{50})}{1000} \times \frac{\frac{1}{2}S_{100}}{S_1 + S_5 + S_{10} + S_{20} + S_{50} + \frac{1}{2}S_{100}} \tag{8}$$

(ii) When the catchment is beyond the simulated 100-year flood-submerged area, then compare the volume of the catchment with the rainfall amount during the 100-year return period. If the former is smaller than the latter, then the catchment capacity of each catchment is calculated as below.

$$\beta = \frac{V}{S} = \frac{\frac{1}{3}\pi r^2 h}{\pi r^2} = \frac{1}{3}h \tag{9}$$

where $\beta$ represents the catchment capacity (m$^3$/ m$^2$), $V$ the maximum rainfall amount (m$^3$) that can be accommodated by a small watershed, $S$ the horizontal plane projection (m$^2$) of a small watershed, $r$ the radius (m) of the underside of a small watershed approximate to a cone, $h$ the height or depth (m) of a small watershed approximate to a cone, and $\pi$ the constant.

When the volume of the catchment was greater than the rainfall amount during the 100-year return period, then the simulation experiment would be repeated. The formula for the simulated flood-submerged area was adopted to calculate the catchment capacity.

### 2.2. Rainwater Storage Capacity Assessment

The rainwater storage capacity refers to the maximum water volume that can be accommodated by a catchment after the depression and infiltration processes during the return periods without causing flood. In other words, the rainwater storage capacity

assessment is made based on the assumption that all the excessive surface rainfall runoff can be infiltrated underground. In this paper, the assessment results of the rainwater storage capacity were expressed as the maximum surface runoff coefficient. The maximum surface runoff coefficient of a catchment with no flood was formulated as below.

$$\varphi_{max} = \frac{HS}{\beta S} = \frac{H}{\beta}$$

(10)

where $\varphi_{max}$ represented the maximum surface runoff coefficient, $H$ the depth of the surface rainfall runoff when flood took place, $S$ the horizontal plane projection area of the catchment, and $\beta$ the catchment capacity. To alleviate urban flood, the comprehensive surface runoff coefficient of each catchment should be smaller than the coefficient when the flood takes place. According to the classification standard of water accumulation degree in the Planning and Design Standard of Drainage and Flood Prevention System in Wuhan (2013), 0.15 m (T < 1 H) is selected as the critical value of no flood to calculate the water-gathering potential and the maximum comprehensive runoff coefficient.

The flood threshold of Wuhan is 0.15 m (within 1 h). The maximum comprehensive surface runoff coefficient of each lot in downtown Wuhan was calculated based on the catchment-capacity results.

*2.3. Ratio of Green Space Calculation*

Different urban lots vary in the underlying surfaces, which have different surface runoff coefficients. These coefficients depend on the green spaces (i.e., the gently-spanning green spaces and the sunken green spaces that retain water) as well as hard roofs and roads. These underlying surfaces totaled over 90% in all lots. Moreover, the surface runoff coefficient of green spaces is about 0.2 while that of the hard roofs and roads is about 0.95. The green space ratio is the ratio between green space and hard ground. The runoff coefficient corresponding to different green space ratio is calculated in Table 1. The minimum green coverage when flood does not take place (i.e., the minimum green coverage at which a lot can give the fullest play of its rainwater storage capacity) can be obtained based on the comprehensive surface runoff coefficient, i.e., the assessment results of rainwater storage capacity, of each lot in Wuhan.

**Table 1.** Correspondence between comprehensive runoff coefficient and green coverage.

| Green Coverage | Runoff Coefficient | Green Coverage | Runoff Coefficient | Green Coverage | Runoff Coefficient |
|---|---|---|---|---|---|
| 95–100 | 0.2–0.24 | 60–65 | 0.46–0.50 | 25–30 | 0.73–0.76 |
| 90–95 | 0.24–0.28 | 55–60 | 0.50–0.54 | 20–25 | 0.76–0.80 |
| 85–90 | 0.28–0.31 | 50–55 | 0.54–0.58 | 15–20 | 0.80–0.84 |
| 80–85 | 0.31–0.35 | 45–50 | 0.58–0.61 | 10–15 | 0.84–0.88 |
| 75–80 | 0.35–0.39 | 40–45 | 0.61–0.65 | 5–10 | 0.88–0.91 |
| 70–75 | 0.39–0.43 | 35–40 | 0.65–0.69 | 0–5 | 0.91–0.95 |
| 65–70 | 0.43–0.46 | 30–35 | 0.69–0.73 | 0 | 0.95 |

## 3. Results and Discussion

*3.1. Experiment and Assessment Results*

A total of 25,894 swales were identified in downtown Wuhan (Figure 7). The depth of each swale equaled the difference between the overflow point and the bottom point (Due to the poor resolution of the urban-space data and the urban-property data, the accuracy of the swale depth was 1 m. The error in the results was significant.) (Figure 8). The simulated results of the 1-year, 5-year, 10-year, 20-year, 50-year, and 100-year flood-submerged area were obtained, with the flood-submerged elevation of each watershed in downtown Wuhan listed in Appendix A Table A3.

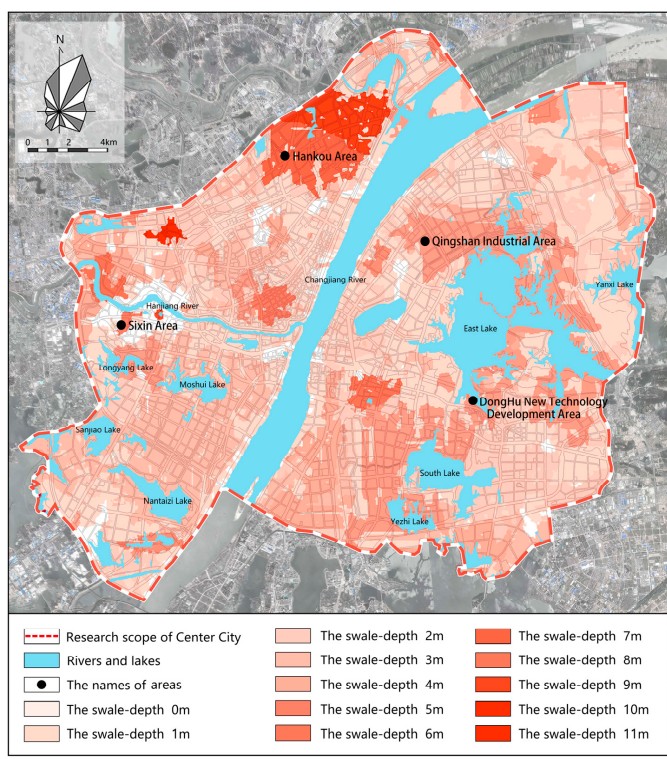

**Figure 7.** Location of downtown Wuhan and flood-submerged area.

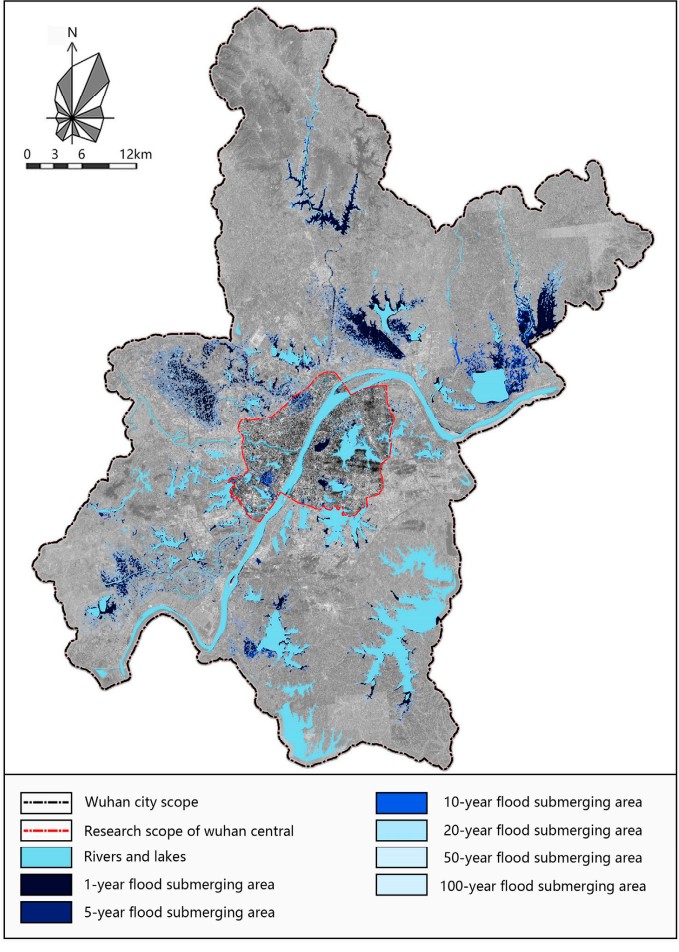

**Figure 8.** Swales in downtown Wuhan.

According to the simulation results, the Nanhu residential area, the Qingshan industrial area, the southeast Donghu high-tech development zone, the Sixin district, and the northwest Hankou suffered severe submerging disasters in all return periods. There were differences between the simulation results and the actual rainfall amount. The simulation experiment did not take the confluence among watersheds into consideration. Therefore, the simulated results for areas with low elevation were smaller than the actual amount while those for areas with high elevation were larger than the actual amount. Based on the actual situation of city, the spatial data were modified in Arcgis to make up for the poor resolution of the urban space data and the urban property data or the shortcomings with data. Nonetheless, errors remained during the simulation process, making the results less accurate and reasonable.

Figure 9 reveals the simulated catchment capacity of the flood-submerged areas during various return periods, Figure 10 reveals the catchment capacity of areas where the catchment capacity of the swales is smaller than the rainfall volume of 100-year return period, and Figure 11 reveals the catchment capacity of areas where the catchment capacity of the swales is greater than the rainfall volume of 100-year return period. The catchment capacity of downtown Wuhan, as shown in Figure 12, is the combination of results of Figures 9 and 10. There were errors when calculating the catchment capacity of swales, which were approximated to the cones. The actual hydrologic processes were not taken into consideration when the simulated catchment capacity of the flood-submerged areas was calculated. The difference in the infiltration speed of various areas would inevitably result in rainfall concentration. For instance, if there is rainfall flowing from $S_5$ to $S_1$, then the actual value of $\beta_5$ is smaller than the calculation value, while the actual value of $\beta_1$ is greater than the calculation value. But these slight errors were neglected in the calculation process, i.e., the infiltration speed of all areas were taken as the same. The infiltration process was taken into consideration in the simulated flood-submerging experiment. The simulated catchment capacity was therefore smaller than the actual value.

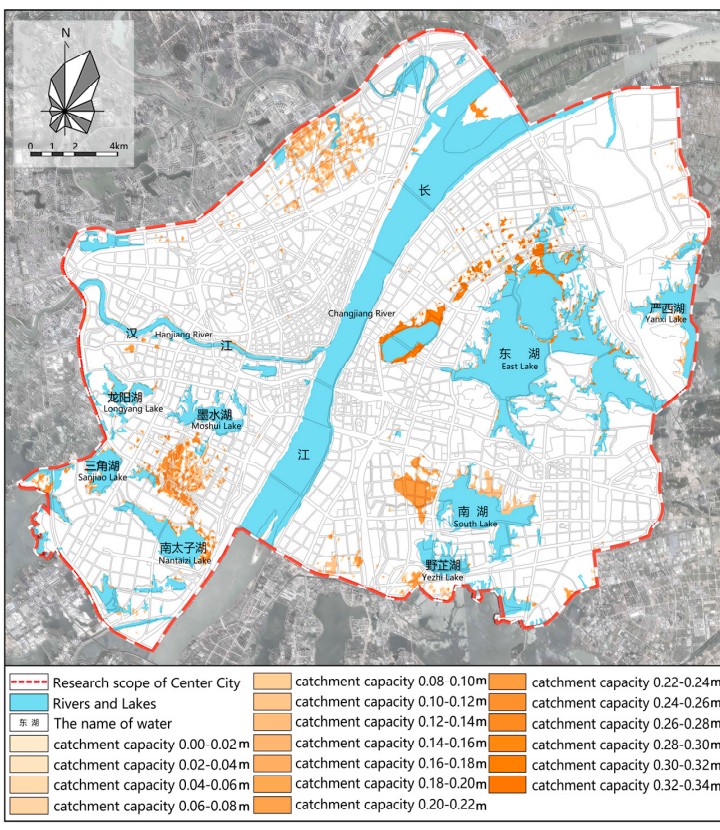

**Figure 9.** Simulated catchment capacity of the flood-submerged areas.

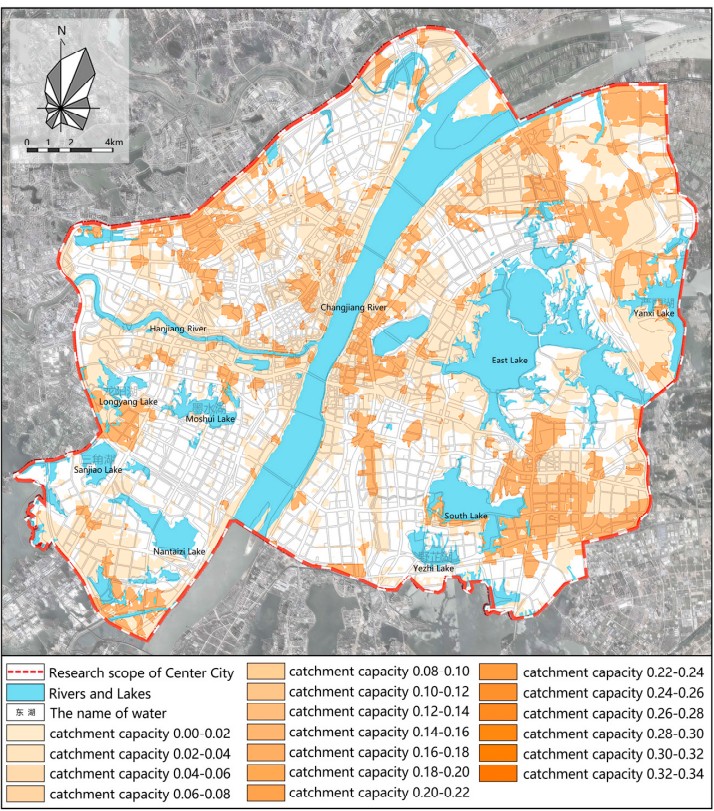

**Figure 10.** Catchment capacity of areas where the catchment capacity of the swales is smaller than the rainfall volume of 100-year return period.

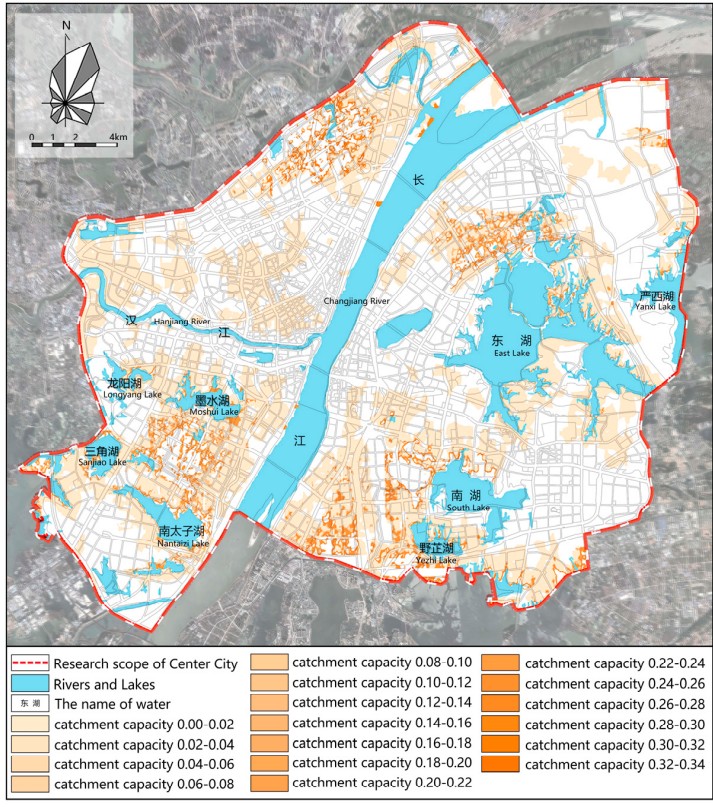

**Figure 11.** Catchment capacity of areas where the catchment capacity of the swales is greater than the rainfall volume of 100-year return period.

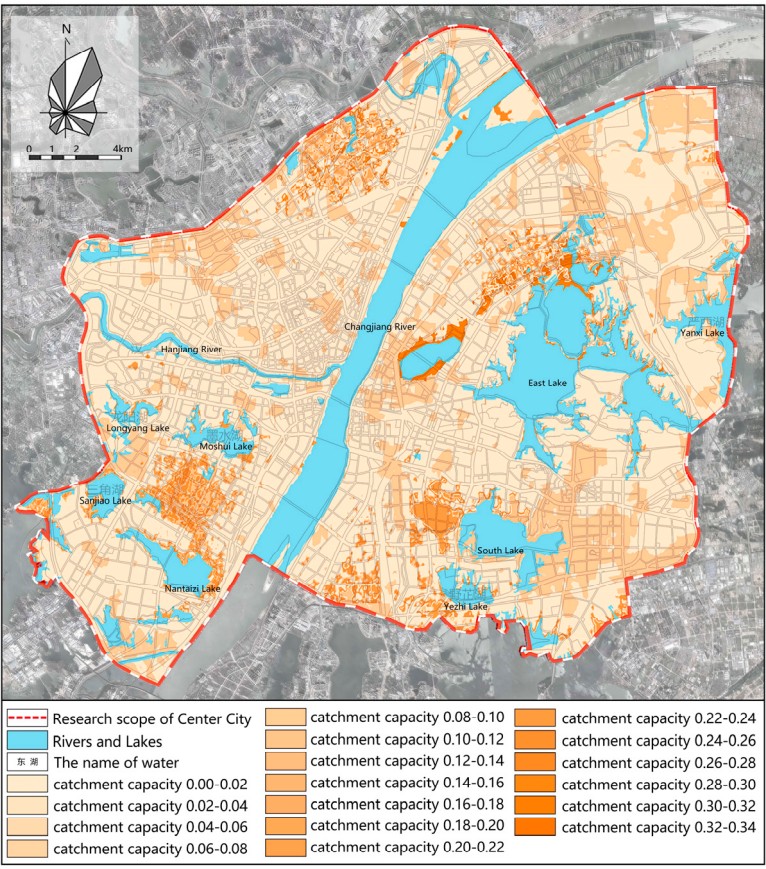

**Figure 12.** Catchment capacity of downtown Wuhan.

The maximum comprehensive surface runoff coefficient under the 100-year return period standard of the lots in downtown Wuhan was obtained based on the catchment-capacity simulation experiment, as shown in Figure 13. The coefficient was viewed as the final evaluation results of the rainwater storage capacity, i.e., the smaller the maximum surface runoff coefficient is, the larger that rainwater storage capacity will be. As discussed before, the comprehensive surface runoff coefficient is functionally correlated with the green coverage. The minimum green coverage under this evaluation standard, i.e., the minimum ratio between the green area and the total area of a lot when the rainwater storage capacity gives the fullest play, would be obtained. The results, together with the other factors, were employed to define the green coverage of the 1530 lots in the Wuhan Comprehensive Plan 2006–2020, as shown in Figure 14. When compiling future urban plans, the results can be used to analyze the green-space distribution characteristics in different land-use types and to adjust the use types accordingly.

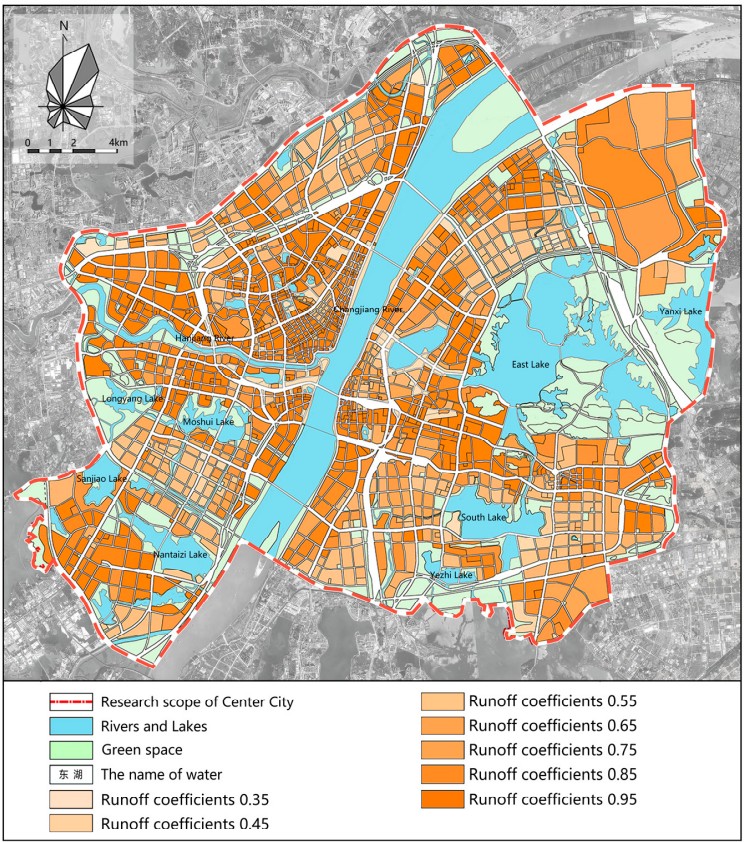

**Figure 13.** Runoff coefficients of downtown Wuhan.

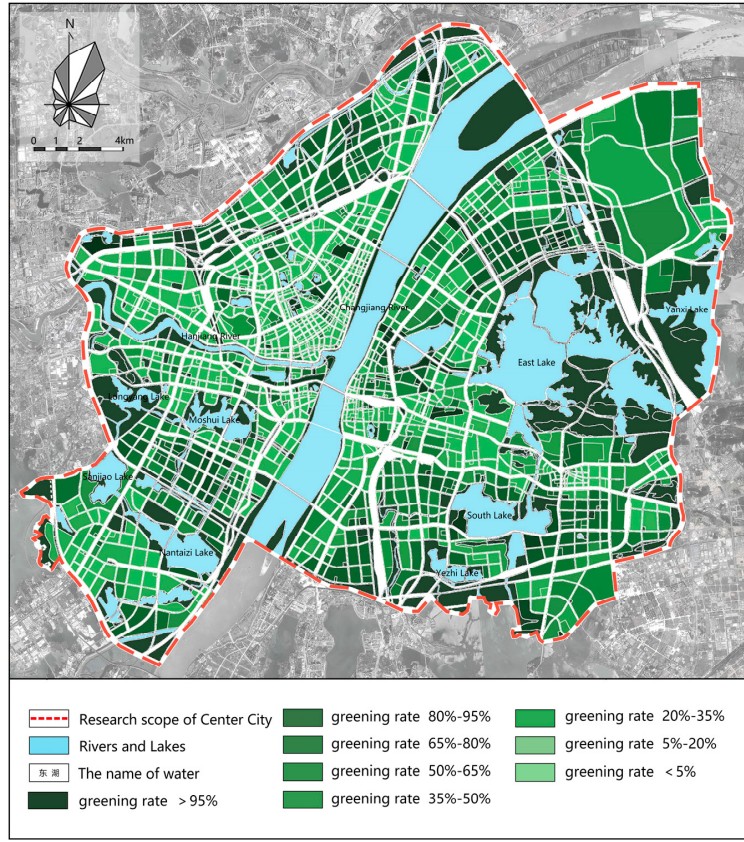

**Figure 14.** Minimum green coverage and green coverage of downtown Wuhan.

*3.2. Experimental Results and Analysis*

Figure 15 is the result of combining the actual flood-submerged areas and the water storage capacity of south downtown Wuhan (On 9 July 2016, Wuhan was hit by the most severe flood over the past three years. The waterlogged areas on this day were obtained by recognizing the water bodies on the satellite image via GIS. Limited by data access, this study recognized and analyzed the water bodies in the south downtown Wuhan). The comparison between the results, satellite images, and field observation revealed three main cases:

① When the rainwater storage capacity was the same, most areas within the flood-submerged areas often had lower green coverage than those outside the flood-submerged areas. This shows that increasing the proportion of green space can effectively reduce urban flood.

② A few areas with greater rainwater storage capacity and green coverage were within the flood-submerged areas because the rainwater collected exceeded the infiltration speed of the green space.

③ There were also areas with greater rainwater storage capacity and smaller green coverage not within the submerged areas or areas with smaller rainwater storage capacity and greater green coverage within the submerged areas. This was caused by experiment errors.

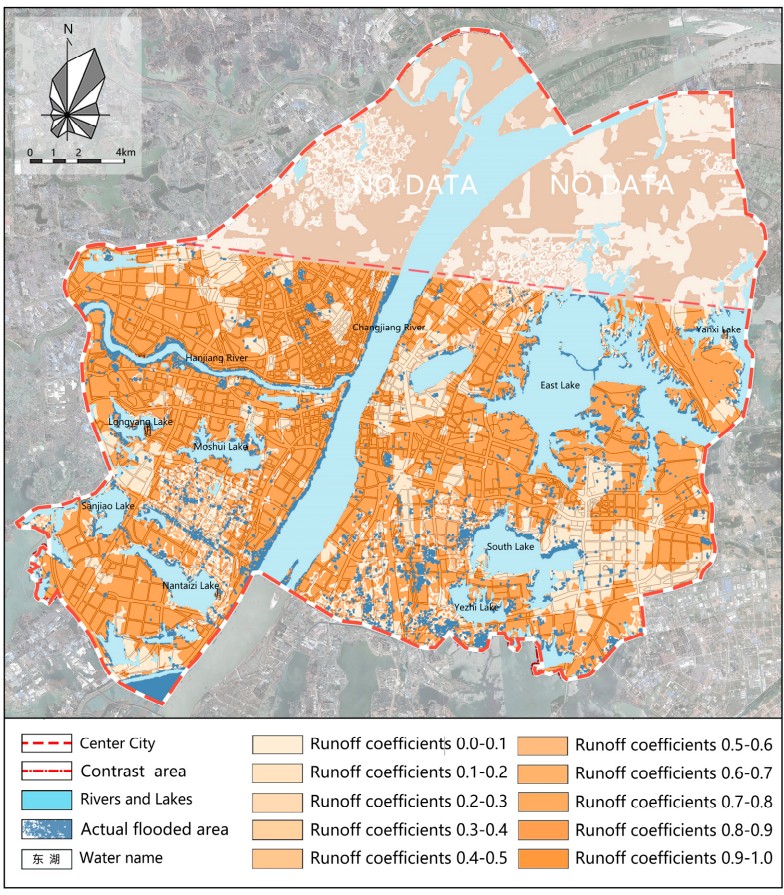

**Figure 15.** Spatial characteristics of the flood-submerged areas in south downtown Wuhan based on the model.

As a result of the rain flood storage evaluation model analysis, for the same land-use type (the smaller the comprehensive runoff coefficient value in the figure is), the larger the stormwater storage potential value is, the larger the proportion of the overlap area between it and the submerged area. In other words, areas with greater rainwater storage capacity

(smaller comprehensive runoff coefficient values in the figure) are more vulnerable to be flooded. The flood-storage evaluation results proved to be useful.

Therefore, compared with SCS-CN submersion simulation results (Figure 16), simulation results from this model have a higher coincidence rate of submerged areas, indicating that results have higher accuracy and smaller errors.

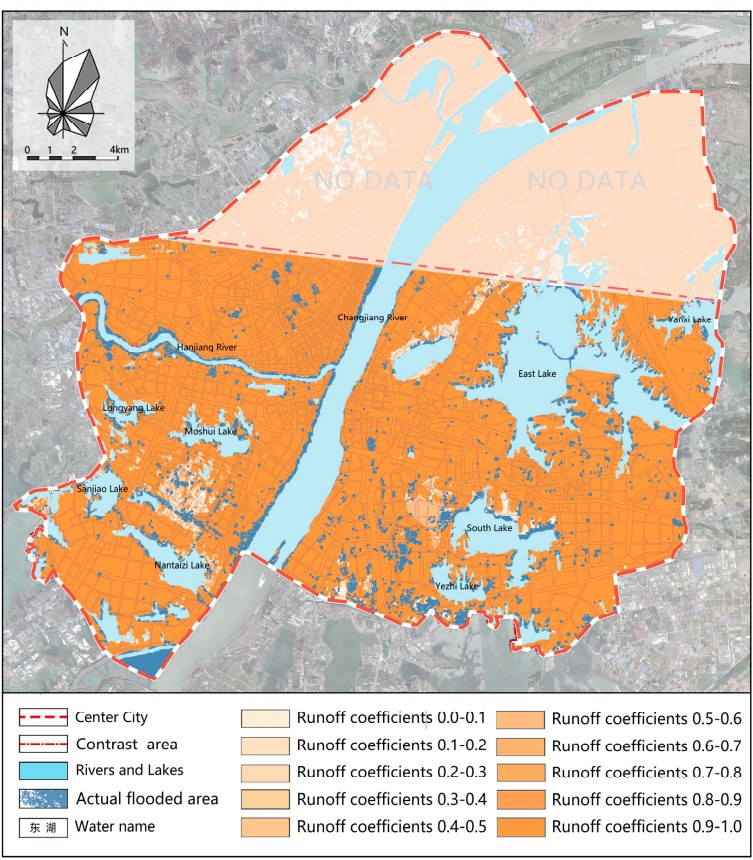

**Figure 16.** Spatial characteristics of the flood-submerged areas in south downtown Wuhan based on the SCS-CN.

## 4. Conclusions

This study designs the rainwater storage capacity evaluation model. Firstly, the SCS-CN model based on hydrologic flood-submerging simulation experiment is improved by developing the swale recognition experiment. The improved results reflect the actual flood-submerged conditions better than the SCS-CN model. Secondly, based on the hydrologic process principles, the model translated the quantitative spatial data obtained from the flood-submerging simulation experiment into the comprehensive surface runoff coefficient and evaluated rainwater storage capacity quantitatively, finally proposing rainwater storage capacity to indicate the responsiveness of the urban flood catchments.

In downtown Wuhan, the rainwater storage capacity evaluation model has great evaluation results. Additionally, the evaluation results are translated into the green coverage, which is applied to determine the land-use types in the Wuhan Comprehensive Plan. Also, in hydrology, the evaluation results are quantitative references for the plan compilation at the current stage.

This study compares the identified urban flood areas in Wuhan on 9 July 2016 with the rainwater storage capacity evaluation results and combines a spatial characteristics analysis of the flood-submerged areas based on the day's satellite images and survey. It was found that the model, compared with the SCS-CN model, had a higher submerged coincidence rate of simulated and actual submerged area, which proves the effectiveness of this model.

**Supplementary Materials:** The following are available online at https://www.mdpi.com/article/10.3390/w13111517/s1, Table S1: Procedure of passive submergence experiment based on SCS-CN model, Table S2: Steps of water collection simulation experiment in small depression.

**Author Contributions:** Conceptualization, Y.L. and Y.Z.; methodology, Y.L. and P.L.; software, P.L.; validation, P.L., and J.Y.; formal analysis, Y.L.; investigation, J.Y.; resources, Y.Z.; data curation, Y.L. and P.L.; writing—original draft preparation, Y.L. and P.L.; writing—review and editing, Y.Z. and L.Y.; visualization, P.L. and Y.L.; supervision, Y.Z.; project administration, Y.Z.; funding acquisition, Y.Z. All authors have read and agreed to the published version of the manuscript.

**Funding:** This research was funded by China's National Natural Science Foundation 2017 project Study on Disaster Relief Landscape Design and Quantitative Control Approach in Response to Urban Flood Mechanism, grant number 51708426.

**Institutional Review Board Statement:** Not applicable.

**Informed Consent Statement:** Not applicable.

**Data Availability Statement:** The study did not report any data.

**Acknowledgments:** This study is a funded component of China's National Natural Science Foundation 2017 project Study on Disaster Relief Landscape Design and Quantitative Control Approach in Response to Urban Flood Mechanism (51708426) and the independent re-search project Study on the Water Collection Capacity of the Water Ecological Infrastructure from the Perspective of Flood Safety: With Dadonghu Area of Wuhan as Study Area of Wuhan University 2018 (2042018kf0250).

**Conflicts of Interest:** The authors declare no conflict of interest.

## Appendix A

**Table A1.** Data sources of water collection capacity experiment.

| Data | Source |
|---|---|
| Digital Elevation Data at a Resolution Rate of GDEMDEM 30 m in Wuhan | Geographical Information Monitoring Cloud Platform (China). |
| Surface Runoff Coefficient CN | National Land Use Type Classification. Please refer to Table A2 for details (China). |
| Rainstorm Intensity on Extreme Days P | 24-h rainfall for the return periods of 1 year, 5 years, 10 years, 20 years, 30 years, 50 years, and 100 years of Wuhan in Wuhan Water Resource Communique 2017 |
| Vector Data on Land Use | The lands are classified into 6 Class I use types (arable land, woodland, grassland, water bodies, construction land, and unused land) and 25 Class II use types (woodland, shrub wood, open forest land, other types of woodland, and grassland with high, medium, and low coverage, etc.) based on the national land use digital products produced by Landsat 30 m remote sensing in accordance with the LUCC classification system established by Li Jiyuan et al. when developing the China LUCC Temporal-Spatial Platform in the 20th Century. |

**Table A2.** Correspondence between land use types and CN.

| Class I | | Class II | | CN |
|---|---|---|---|---|
| **No.** | **Type** | **No.** | **Type** | |
| 1 | Arable land | 11 | Rice field | 1 |
| | | 12 | Dry land | 0.15 |
| 2 | Woodland | 21 | Forested land | 0.1 |
| | | 22 | Shrub wood | 0.1 |
| | | 23 | Open forest land | 0.1 |
| | | 24 | Others | 0.1 |

**Table A2.** *Cont.*

| Class I | | Class II | | CN |
|---|---|---|---|---|
| No. | Type | No. | Type | |
| | | 31 | Grassland with large coverage | 0.2 |
| 3 | Grassland | 32 | Grassland with medium coverage | 0.2 |
| | | 33 | Grassland with small coverage | 0.2 |
| | | 41 | Rivers and canals | 1 |
| | | 42 | Lakes | 1 |
| 4 | Water bodies | 43 | Reservoirs, ponds, and pools | 1 |
| | | 44 | Permanent glacier and snow | |
| | | 45 | Mud flats | 1 |
| | | 46 | Bottomland | 1 |
| | Urban-rural construction land, industrial and mining land, and residential land | 51 | Urban land | 0.75 |
| 5 | | 52 | Rural settlements | 0.4 |
| | | 53 | Construction land for other purposes | 0.6 |
| | | 61 | Sand | 0.25 |
| | | 62 | Gobi | 0.25 |
| | | 63 | Saline-alkali land | 0.25 |
| 6 | Unused land | 64 | Wetland | 1 |
| | | 65 | Barren earth | 0.25 |
| | | 66 | Barren rock surface | 0.6 |
| | | 67 | Others | |

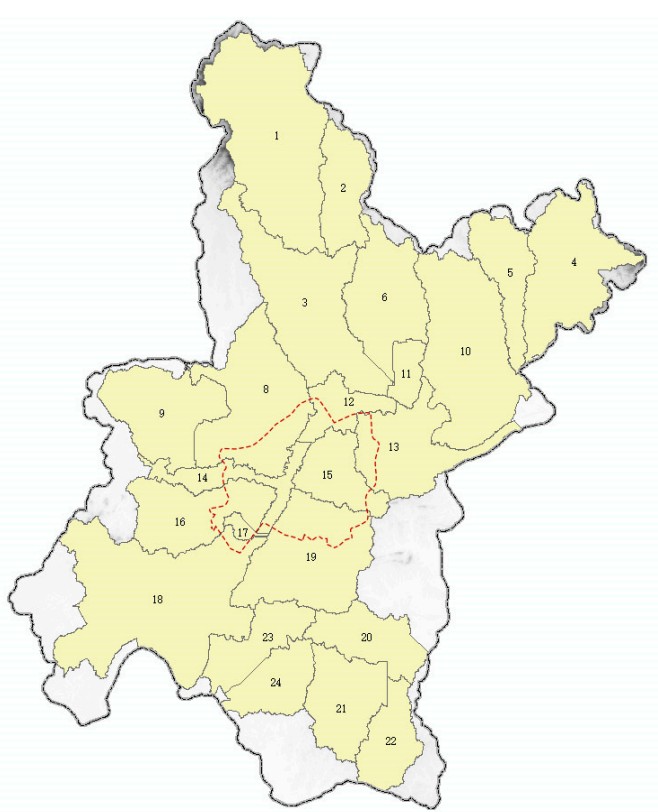

**Figure A1.** Simulation results and serial map of watershed division and submergence in Wuhan city.

Table A3. List of results of SCS-CN model.

| No. | Type | 1-Year Return Period | 5-Year Return Period | 10-Year Return Period | 20-Year Return Period | 50-Year Return Period | 100-Year Return Period |
|---|---|---|---|---|---|---|---|
| | P (mm) | 95 | 162 | 205 | 249 | 303 | 344 |
| | CN | 81.519 | 81.519 | 81.519 | 81.519 | 81.519 | 81.519 |
| 8 | S (mm) | 57.5838 | 57.5838 | 57.5838 | 57.5838 | 57.5838 | 57.5838 |
| | Q (mm) | 92.7348 | 159.7191 | 202.7144 | 246.7113 | 300.7087 | 341.7073 |
| | Drainage area (m$^2$) | 477,686,464 | 477,686,464 | 477,686,464 | 477,686,464 | 477,686,464 | 477,686,464 |
| | Flood volume (m$^3$) | 44,298,165 | 76,295,689 | 96,833,966 | 117,850,674 | 143,644,497 | 163,228,957 |
| | P (mm) | 95 | 162 | 205 | 249 | 303 | 344 |
| | CN | 63.8694 | 63.8694 | 63.8694 | 63.8694 | 63.8694 | 63.8694 |
| 12 | S (mm) | 143.6865 | 143.6865 | 143.6865 | 143.6865 | 143.6865 | 143.6865 |
| | Q (mm) | 89.4846 | 156.3908 | 199.3623 | 243.3432 | 297.3273 | 338.3184 |
| | Drainage area (m$^2$) | 81,160,950 | 81,160,950 | 81,160,950 | 81,160,950 | 81,160,950 | 81,160,950 |
| | Flood volume (m$^3$) | 7,262,659 | 12,692,831 | 16,180,440 | 19,749,971 | 24,131,366 | 27,458,250 |
| | P (mm) | 95 | 162 | 205 | 249 | 303 | 344 |
| | CN | 67.2196 | 67.2196 | 67.2196 | 67.2196 | 67.2196 | 67.2196 |
| 13 | S (mm) | 123.8659 | 123.8659 | 123.8659 | 123.8659 | 123.8659 | 123.8659 |
| | Q (mm) | 90.2187 | 157.1484 | 200.1272 | 244.1129 | 298.1010 | 339.0944 |
| | Drainage area (m$^2$) | 278,770,265 | 278,770,265 | 278,770,265 | 278,770,265 | 278,770,265 | 278,770,265 |
| | Flood volume (m$^3$) | 25,150,312 | 43,808,326 | 55,789,512 | 68,051,425 | 83,101,699 | 94,529,448 |
| | P(mm) | 95 | 162 | 205 | 249 | 303 | 344 |
| | CN | 73.4135 | 73.4135 | 73.4135 | 73.4135 | 73.4135 | 73.4135 |
| 14 | S (mm) | 91.9854 | 91.9854 | 91.9854 | 91.9854 | 91.9854 | 91.9854 |
| | Q (mm) | 91.4170 | 158.3777 | 201.3659 | 245.3579 | 299.3513 | 340.3477 |
| | Drainage area (m$^2$) | 100,142,226 | 100,142,226 | 100,142,226 | 100,142,226 | 100,142,226 | 100,142,226 |
| | Flood volume (m$^3$) | 9,154,707 | 15,860,300 | 20,165,229 | 24,570,693 | 29,977,711 | 34,083,178 |

**Table A3.** *Cont.*

| No. | Type | 1-Year Return Period | 5-Year Return Period | 10-Year Return Period | 20-Year Return Period | 50-Year Return Period | 100-Year Return Period |
|---|---|---|---|---|---|---|---|
| 15 | P (mm) | 95 | 162 | 205 | 249 | 303 | 344 |
| | CN | 73.0794 | 73.0794 | 73.0794 | 73.0794 | 73.0794 | 73.0794 |
| | S (mm) | 93.5671 | 93.5671 | 93.5671 | 93.5671 | 93.5671 | 93.5671 |
| | Q (mm) | 91.3570 | 158.3164 | 201.3041 | 245.2959 | 299.2891 | 340.2853 |
| | Drainage area (m$^2$) | 182,403,969 | 182,403,969 | 182,403,969 | 182,403,969 | 182,403,969 | 182,403,969 |
| | Flood volume (m$^3$) | 16,663,895 | 28,877,548 | 36,718,684 | 44,742,962 | 54,591,529 | 62,069,405 |
| 16 | P (mm) | 95 | 162 | 205 | 249 | 303 | 344 |
| | CN | 78.8657 | 78.8657 | 78.8657 | 78.8657 | 78.8657 | 78.8657 |
| | S (mm) | 68.0665 | 68.0665 | 68.0665 | 68.0665 | 68.0665 | 68.06650 |
| | Q (mm) | 92.3305 | 159.3087 | 202.3022 | 246.2978 | 300.2942 | 341.2922 |
| | Drainage area (m$^2$) | 293,267,090 | 293,267,090 | 293,267,090 | 293,267,090 | 293,267,090 | 293,267,090 |
| | Flood volume (m$^3$) | 27,077,500 | 46,720,017 | 59,328,586 | 72,231,058 | 88,066,414 | 100,089,778 |
| 17 | P (mm) | 95 | 162 | 205 | 249 | | 344 |
| | CN | 81.3199 | 81.3199 | 81.3199 | 81.3199 | | 81.3199 |
| | S (mm) | 58.3466 | 58.3466 | 58.3466 | 58.3466 | | 58.3466 |
| | Q (mm) | 92.7053 | 159.6892 | 202.6844 | 246.6812 | | 341.6770 |
| | Drainage area (m$^2$) | 19,719,894 | 19,719,894 | 19,719,894 | 19,719,894 | | 19,719,894 |
| | Flood volume (m$^3$) | 1,828,138 | 3,149,055 | 3,996,915 | 4,864,527 | 0 | 6,737,835 |
| 18 | P (mm) | 95 | 162 | 205 | 249 | | 344 |
| | CN | 69.8384 | 69.8384 | 69.8384 | 69.8384 | | 69.8384 |
| | S (mm) | 109.6967 | 109.6967 | 109.6967 | 109.6967 | | 109.6967 |
| | Q (mm) | 90.74866 | 157.6931 | 200.6764 | 244.6652 | | 339.6506 |
| | Drainage area (m$^2$) | 748,159,926 | 748,159,926 | 748159926 | 748,159,926 | | 748,159,926 |
| | Flood volume (m$^3$) | 67,894,515 | 117,979,732 | 150,138,065 | 183,048,699 | 0 | 254,113,019 |

**Table A3.** *Cont.*

| No. | Type | 1-Year Return Period | 5-Year Return Period | 10-Year Return Period | 20-Year Return Period | 50-Year Return Period | 100-Year Return Period |
|-----|------|---------------------|---------------------|----------------------|----------------------|----------------------|------------------------|
|     | P (mm) | 95 | 162 | 205 | 249 | | 344 |
|     | CN | 73.6103 | 73.6103 | 73.6103 | 73.6103 | | 73.6103 |
|     | S (mm) | 91.0604 | 91.0604 | 91.0604 | 91.0604 | | 91.0604 |
| 19  | Q (mm) | 91.4521 | 158.4136 | 201.4020 | 245.3942 | | 340.3841 |
|     | Drainage area (m$^2$) | 453,176,540 | 453,176,540 | 453,176,540 | 453,176,540 | | 453,176,540 |
|     | Flood volume (m$^3$) | 41,443,968 | 71,789,334 | 91,270,661 | 111,206,907 | 0 | 154,254,124 |

**Table A4.** A list of simulated submergence elevation of SCS-CN.

| No. | 1-Year Return Period | | | 5-Year Return Period | | | 10-Year Return Period | | |
|-----|---------------------|---|---|---------------------|---|---|----------------------|---|---|
|     | Surface Runoff (Mm) | Flood Volume (M$^3$) | Flood Elevation (M) | Surface Runoff (Mm) | Flood Volume (M$^3$) | Flood Elevation (M) | Surface Runoff(Mm) | Flood Volume (M$^3$) | Flood Elevation (M) |
| 8  | 92.735 | 44,298,165,157 | 18.683 | 159.719 | 76,295,689,051 | 19.339 | 202.714 | 96,833,966.441 | 19.690 |
| 12 | 89.485 | 7,262,659,118 | 18.641 | 156.391 | 12,692,831,670 | 19.734 | 199.362 | 16,180,440.776 | 20.381 |
| 13 | 90.219 | 25,150,312,969 | 18.541 | 157.148 | 43,808,326,252 | 19.165 | 200.127 | 55,789,512.965 | 19.528 |
| 14 | 91.417 | 9,154,707,467 | 17.902 | 158.378 | 15,860,300,683 | 19.089 | 201.366 | 20,165,229.782 | 19.709 |
| 15 | 91.357 | 16,663,895,540 | 20.088 | 158.316 | 28,877,548,772 | 20.436 | 201.304 | 36,718,684.104 | 20.658 |
| 16 | 92.331 | 27,077,500,567 | 19.301 | 159.309 | 46,720,017,924 | 19.827 | 202.302 | 59,328,586.716 | 20.119 |
| 17 | 92.705 | 1,828,138,883 | 18.707 | 159.689 | 3,149,055,339 | 19.233 | 202.684 | 3,996,915.852 | 19.550 |
| 18 | 90.749 | 67,894,515,627 | 20.526 | 157.693 | 117,979,732,158 | 21.221 | 200.676 | 150,138,065.625 | 21.549 |
| 19 | 91.452 | 41,443,968,539 | 18.74 | 158.414 | 71,789,334,635 | 19.238 | 201.402 | 91,270,661.982 | 19.511 |

**Table A4.** *Cont.*

| No. | 20-Year Return Period | | | 50-Year Return Period | | | 100-Year Return Period | | |
|---|---|---|---|---|---|---|---|---|---|
| | Surface Runoff (mm) | Flood Volume (m³) | Flood Elevation (m) | Surface Runoff (mm) | Flood Volume (m³) | Flood Elevation (m) | Surface Runoff (mm) | Flood Volume (m³) | Flood Elevation (m) |
| 8 | 246.711 | 11,785,0674,343 | 20.017 | 300.709 | 143,644,497,809 | 20.315 | 341.707 | 163,228,957,129 | 20.538 |
| 12 | 243.343 | 19,749,971,050 | 20.97 | 297.327 | 24,131,366,380 | 21.511 | 338.318 | 27,458,250,099 | 21.89 |
| 13 | 244.113 | 68,051,425,946 | 19.886 | 298.101 | 83,101,699,609 | 20.274 | 339.094 | 94,529,448,899 | 20.556 |
| 14 | 245.358 | 24,570,693,624 | 20.237 | 299.351 | 29,977,711,937 | 20.795 | 340.348 | 34,083,178,465 | 21.133 |
| 15 | 245.296 | 44,742,962,683 | 20.881 | 299.289 | 54,591,529,940 | 21.133 | 340.285 | 62,069,405,427 | 21.317 |
| 16 | 246.298 | 72,231,058,133 | 20.385 | 300.294 | 88,066,414,701 | 20.706 | 341.292 | 100,089,778,212 | 20.942 |
| 17 | 246.681 | 4,864,527,785 | 19.866 | 300.679 | 5,929,349,292 | 20.217 | 341.677 | 6,737,835,900 | 20.471 |
| 18 | 244.665 | 183,048,699,778 | 21.866 | 298.656 | 223,442,326,941 | 22.155 | 339.651 | 254,113,019,597 | 22.342 |
| 19 | 245.394 | 11,120,6907,245 | 19.791 | 299.388 | 135675504,457 | 20.101 | 340.384 | 154,254,124,751 | 20.324 |

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
