# Peer review of "Green Space Optimization Strategy to Prevent Urban Flood Risk in the City Centre of Wuhan"

_water, doi:10.3390/w13111517_

Round 1
Reviewer 1 Report
The Title - matches the content
Summary - synthetically reflects the substantive content of the manuscript. However, the synthetic description of the obtained results differs from the conclusions presented in the final conclusions. (Conclusions need to be improved).
The introduction presents commonly known problems. The authors based mainly on articles published before 2017 (14 out of 22 items).
It should be noted that in recent years, a number of important practical and cognitive studies have been carried out on the topic undertaken by the authors of the research. I suggest making a broader literature review in order to update the state of knowledge on the subject matter.
Additionally, in my opinion, there is no clearly described research goal.
I suggest to improve the final part of the section: (lines 103-109) to indicate the clear purpose of the article, e.g.
For example:
The aim of the article is to present the results of research on the modification of the SCS-CN model to improve the obtained results in the field of .....
Materials and methods
In the line : 113 – „GIS hydrological analysis tools…” it be described in more detail.
Model SCS-CN
The description of the model should be extended. Alternatively, a reference to the source material should be provided.
The methodology used is quite simple, but aimed at achieving the goal assumed by the authors.
Conclusions:
Conclusions should be improve. The first part of the section summarizes what was done in the article in a rather chaotic fashion. Only one accurate observation of the authors can be drawn from the presented conclusions:
"The results suggest that areas with greater rainwater storage capacity are more prone to being flooded and that the incidence of waterlogging is directly related to the greening rate."
I suggest the authors consider an in-depth study of the article from the point of view of drawing more scientifically relevant conclusions. For example, by capturing the results obtained from the method refined by the authors with the results obtained from the hydrodynamic modeling software.
References:
There is no excessive number of self-citations
22 references
Only 8 publications are less than 5 years old. It is suggested to broaden the literature review towards recently published studies in this field.
Author Response
Dear editors and experts:
Thank you very much for your valuable comments on this study. This has contributed a lot to our research. Based on the suggestions, we have made the adjustments to this article. Please find it in attachment.

Reviewer 2 Report
Dear Authors, I have included my comments and suggestions in the document herein attached.

Author Response
Dear editors and experts:
Thank you very much for your valuable comments on this study. This has contributed a lot to our research. Based on the suggestions, we have made adjustments to this article. Please find in attachment.

Round 2
Reviewer 1 Report
The manuscript has been improved.
In my opinion, the article can be considered for publication by the editor.
Author Response
Thank you very much for your comments, which will be of great help to our future research.
Reviewer 2 Report
Dear Authors,
I am happy to see that most of my observations were accomplished. However, I think that some further efforts should be done to improve the manuscript.
My main concern is about the English. As revised, the English style needs to be still enhanced. For instance, lines 343-345 and 378-284 are characterized by a very poor English language style.
I think that the reference “Crispino, Gaetano; Pfister, Michael; Gisonni, Corrado. (2019) Supercritical flow in junction manholes under invert- and obvert-aligned set-ups, Journal of Hydraulic Research, 57:4, 534-546, DOI: 10.1080/00221686.2018.1494056” should be added within the additional references (lines 87-89) dealing with the failure of the urban drainage system as a possible cause of the urban flood.
As minor comments, please be careful when you read the text because some typos are still present (See Eq. S10 where the symbols are not correctly typeset). In some lines the word "pouring point" is still used, despite I highlighted that its utilization is quite uncommon in a scientific paper.
The sentence "The flow threshold was set at 1,500,000" continues to be quite unclear in my opinion. What is a flow threshold? Flow is usually used as a quantitative concept (discharge). Contrarily, you used it as a not-dimensional quantity. Moreover, may you quantify the error encountered by approximating the swale capacities with the cones? I already asked but you did not reply to this comment.
Finally, Abstract is still too lengthy.
Author Response
Dear expert and editor,
Thanks for your advice, which is very helpful for our research.
We’re sorry that we didn't express the problems clearly last time, which caused misunderstanding. Now supplemented as follows:
- Expert:Abstract is still too lengthy.
Respond:According to suggestion, we have condensed it.
- Expert:I think that the reference “Crispino, Gaetano; Pfister, Michael; Gisonni, Corrado. (2019) Supercritical flow in junction manholes under invert- and obvert-aligned set-ups, Journal of Hydraulic Research, 57:4, 534-546, DOI: 10.1080/00221686.2018.1494056” should be added within the additional references (lines 87-89) dealing with the failure of the urban drainage system as a possible cause of the urban flood.
Respond:We have read the literature and added to the reference.
- Expert:The sentence "The flow threshold was set at 1,500,000" continues to be quite unclear in my opinion. What is a flow threshold? Flow is usually used as a quantitative concept (discharge). Contrarily, you used it as a not-dimensional quantity.
Respond:1500000 is the parameter value required when using the Fow Direction Tools and the Flow Tools in the process of GIS watershed division. According to the GIS tips, this value is related to the number of grids in GIS and does not represent the flow value.
In fact, the smaller the input value, the more the number of watershed, the greater the calculation workload, and the more accurate the experimental results. On the contrary, the larger the input value, the fewer the number of watershed, the more convenient calculation.This needs to be balanced according to actual needs.
After many experiments, the input value of 1500000 is more appropriate. On the one hand, the 24 watershed values obtained are relatively close to the real inundation; on the other hand, the purpose of this experiment can be satisfied with a relatively small amount of work.
- Expert:may you quantify the error encountered by approximating the swale capacities with the cones?
Respond:Due to the urban macro scale, the real topography of the concave and convex quantity is temporarily unknown,so the error encountered by approximating the swale capacities with the cones can’t be quantified. However, we still believe that this suggestion is valuable. In the future, we can try to use terrain indexes such as roughness factor to roughly estimate the error, but this needs more time and more data to support.
In addition, thanks for your advices about English expression. We have corrected the grammar of the relevant paragraphs, and modified the whole article:
- Expert:My main concern is about the English. As revised, the English style needs to be still enhanced. For instance, lines 343-345 and 378-284 are characterized by a very poor English language style.
Respond:Thanks for your reminding. We have modified the lines 243-245, 343-345 and 378-384, and improved the English expression of the whole text.
- Expert:As minor comments, please be careful when you read the text because some typos are still present (See Eq. S10 where the symbols are not correctly typeset).
Respond:It would be digital gibberish. We fixed the garbled and checked for other similar problems.
- Expert:In some lines the word "pouring point" is still used, despite I highlighted that its utilization is quite uncommon in a scientific paper.
Respond:We replace the “pouring point” with “overflow point”. The overflow point refers to the lowest point on the upper edge of each depression, beyond which the water in the depression will overflow.
- Expert:The sentence "The flow threshold was set at 1,500,000" continues to be quite unclear in my opinion. What is a flow threshold?
Respond:"Flow Threshold" is the name of an input value in the GIS.
Thank you again for kind advice, which was of great help to us. If you have any questions, please discuss them further.
Best wishes!
Yan Zhou;Yajing Liu;Pengchen Li;
Liuqi Yang;Jianing Yu
2021年05月23日